

# Transcriptomic insights into grain size development in naked barley (*Hordeum vulgare* L. var. *nudum* Hook. f): based on weighted gene co-expression network analysis

Yan Wang[1,2,3,4], Jiahao Zhou[5], Mingqi Yang[5], Youhua Yao[1,2,3,4], Yongmei Cui[1,2,3,4], Xin Li[1,2,3,4], Baojun Ding[1,2,3,4], Xiaohua Yao[1,2,3,4] and Kunlun Wu[2,3,4,5]

[1] Qinghai Academy of Agricultural and Forestry Sciences, Qinghai University, Xining, Qinghai, China
[2] Laboratory for Research and Utilization of Qinghai Tibet Plateau Germplasm Resources, Xining, Qinghai, China
[3] Qinghai Subcenter of National Hulless Barley Improvement, Xining, Qinghai, China
[4] Qinghai Key Laboratory of Hulless Barley Genetics and Breeding, Xining, Qinghai, China
[5] College of Agriculture and Animal Husbandry, Qinghai University, Xining, Qinghai, China

Corresponding authors
Xiaohua Yao,
yaoxiaohua009@126.com
Kunlun Wu, wklqaaf@163.com

## ABSTRACT

**Background:** This study investigated the molecular mechanisms underlying grain size variation between two distinct naked barley varieties using comprehensive phenotypic and transcriptomic (RNA-Seq) analyses.

**Methods:** In this study, we employed a comparative transcriptomics approach to analyze two naked barley varieties: the large-grained Shenglibai and the small-grained Lalu Qingke. Our investigation focused on three critical developmental periods of grain growth (early, mid, and late grain-filling periods). By integrating longitudinal three-dimensional phenotypic data with temporal expression profiles and applying weighted gene co-expression network analysis (WGCNA), we successfully identified gene modules that co-vary with morphological expansion.

**Results:** Phenotypic assessments revealed that grains underwent rapid expansion during the filling period, with significant differences in grain width (GW) and thickness (GT) across all three developmental periods. In contrast, grain length (GL) remained relatively consistent by the end of the filling period. Transcriptome sequencing identified a peak in differentially expressed genes (DEGs) during the mid-filling period, indicating that the regulation of grain size development is most active in the early and mid-filling phases. WGCNA identified a blue module strongly correlated with grain size, which was significantly enriched in key metabolic pathways, including starch and sucrose metabolism. Further analysis identified seven hub genes, among which *HvENO1* exhibited pronounced upregulation in large-grain varieties during the mid-to-late filling periods, closely aligning with the observed phenotypic traits. Real-time quantitative reverse transcription polymerase chain reaction (qRT-PCR) validation confirmed the period-specific and variety-specific expression patterns of these genes, further supporting the potential of these genes as targets for improving grain size in breeding.

# INTRODUCTION

Grain crops are of immense nutritional and industrial importance, forming the cornerstone of staple diets in most countries. Among these, barley (*Hordeum vulgare* L.) stands as one of the oldest cultivated cereal crops worldwide, surpassed only by wheat, rice, and maize (*El-Hashash & El-Absy, 2019*; *Güngör et al., 2024*). Barley is categorized into two-row and six-row varieties, with the six-row naked barley, known as 'Qingke' in Chinese, predominantly grown in the high-altitude regions of Tibet, Qinghai, Gansu, Sichuan, Yunnan, and other parts of China (*Wang et al., 2024*; *Shahmoradi, 2021*; *Zeng et al., 2018*; *Guo et al., 2020*). Notably, barley is an excellent source of plant fiber and protein, with its soluble dietary fiber and total fiber content surpassing that of other cereal crops (*Guo et al., 2020*; *Farag, Xiao & Abdallah, 2022*; *Kumar, Narwal & Verma, 2022*). On the Tibetan Plateau, naked barley is not only the primary food crop but also deeply embedded in the local culture and diet, as evidenced by traditional foods such as barley wine and tsampa (*Wang et al., 2023*). The development of genetically stable, large-grained naked barley varieties suitable for alpine farming regions is critical for ensuring food security and promoting economic growth among farmers and herdsmen. Moreover, the widening gap between naked barley supply and demand highlights the pressing need to enhance naked barley production, improve yield, and elevate its quality.

In model plants such as rice (*Oryza sativa* L.) and *Arabidopsis thaliana*, numerous quantitative trait loci (QTLs)/genes regulating grain size have been cloned, and multiple signaling pathways controlling grain size have been identified. These primarily include the ubiquitin-proteasome pathway, G-protein signaling pathway, mitogen-activated protein kinase (MAPK) pathway, phytohormone pathway, transcription factor pathway, and IKU signaling pathway, among others (*Li, Xu & Li, 2019*; *Ren et al., 2019*; *Jiang et al., 2022*). To date, research has identified several regulatory mechanisms and candidate genes associated with hulled barley grain size. Key genes controlling spike morphology (*e.g.*, *Vrs* genes), caryopsis covering (*e.g.*, *Nud* gene), and plant height (*e.g.*, *sdw/denso* genes) have been characterized (*Youssef et al., 2017*; *Koppolu et al., 2013*; *Bull et al., 2017*; *Taketa et al., 2008*). However, there are few reports on the genetic patterns and regulatory mechanisms of large-grained naked barley under the unique environmental conditions of the Tibetan Plateau. Although previous studies have identified candidate genes associated with hulled barley, these were primarily investigated in hulled barley, and whether these genes have similar functions and regulatory mechanisms in naked barley remains unexplored.

We implemented a comparative transcriptomics strategy using two naked barley accessions with extreme grain phenotypes: The large-grained variety Shenglibai and the small-grained variety Lalu Qingke. Kernel development was divided into three critical stages representing distinct phases of dimensional determination: the initial cell proliferation phase (early filling period), the expansion period (mid-filling period), and the maturation period (late filling period). We performed weighted gene co-expression

network analysis (WGCNA) analysis by integrating phenotypic data (grain length, GL; grain width, GW; grain thickness, GT) of grains from three developmental periods with RNA sequencing (RNA-seq) gene expression profiles to identify key co-expression modules associated with grain size. Hub genes from the most significant modules were selected based on their degree values, and critical grain size-related genes were ultimately determined by combining differential expression patterns from RNA-seq data with functional gene annotations. This multilayered approach not only identifies candidate regulators of high-altitude grain enlargement but also establishes a molecular framework for understanding how Tibetan naked barley reconciles vigorous grain filling with environmental stress resilience–a critical adaptation missing in lowland barley cultivars. The gene repertoire provides genetic targets for naked barley improvement while elucidating the molecular basis of grain trait adaptation to high-altitude selection from an evolutionary genetics perspective.

## MATERIALS AND METHODS

### Plant materials

The large- and small-grain naked barley varieties Shenglibai and Lalu Qingke, respectively, were sourced from the National Crop Germplasm Bank (Beijing, China). They were planted in April 2024 at the experimental base of the Crop Research Institute of Qinghai Academy of Agriculture and Forestry (36.73°N, 101.75°E). Grains of the two varieties at early, mid and late filling periods were collected for gene expression analysis. Six biological replicates were used for each sample, three of which were used for determining grain phenotypes and the other three for grain transcriptome sequencing. The sampled grains were immediately frozen with liquid nitrogen and stored in a −80 °C refrigerator.

### Measurement of grain size

Two hundred intact grains were randomly selected from each developmental period of the two varieties, and images were collected and analyzed for GL, GW, and GT. GL, GW and GT were measured by Wanshen SC-G automatic grain testing analyser (Hangzhou Wanshen Testing Technology Co., Ltd., Hangzhou, China).

### RNA extraction, library construction and sequencing

Total RNA was extracted using TRIzol reagent according to the instructions, and RNA purity and quantification were determined using a NanoDrop 2000 spectrophotometer (Thermo Fisher Scientific, Waltham, MA, USA), and RNA integrity was assessed using an Agilent 2100 Bioanalyzer (Agilent Technologies, Santa Clara, CA, USA) to assess RNA integrity. Transcriptome libraries were constructed using the VAHTS Universal V5 RNA-seq Library Prep kit according to the instructions. The libraries were sequenced using the llumina Novaseq 6000 sequencing platform and 150 bp bipartite reads were generated. Raw reads in FASTQ format were processed using FASTP software (*Chen et al., 2018*), and clean reads were obtained after removing low-quality reads for subsequent data analysis. Transcriptome sequencing was performed by oebiotech biotechnology

(Shanghai, China). Transcriptome analysis was performed by Shanghai Meiji Biomedical Technology Co (Shanghai China).

## Gene expression analysis and differential gene screening

In RNA-Seq analysis, the expression level of a gene was determined by quantifying the number of sequence reads mapped to its genomic region, referred to as read counts. To facilitate a detailed assessment of gene and transcript expression, the software RSEM (http://deweylab.github.io/RSEM/) was employed to quantify expression levels separately for genes and transcripts.

After obtaining the read counts number of the genes, the multi-sample (>2) project was analysed for differential expression of genes between samples to identify genes that were differentially expressed between samples, and then to study the function of the differentially expressed genes. The software used for differential expression was DESeq2 (http://bioconductor.org/packages/stats/bioc/DESeq2/), and the screening criteria for differentially expressed genes were: False discovery rate (FDR) < 0.05 and $|Log2FC| \geq 1$. When a gene met both conditions, it was regarded as a differentially expressed gene (DEG). FoldChange indicates the ratio of expression between two samples (groups). FDR was obtained by correcting the $P$-value of the significance of difference, which indicates the significance of difference. To facilitate comparison, the multiplicity of differences is taken as logarithmic value and expressed as log2FC, the larger the absolute value of log2FC, the smaller the FDR value of the genes in the two groups of samples, the more significant the change of differences.

## Real-time quantitative reverse transcription polymerase chain reaction analysis

cDNA synthesis was done using TaKaRa PrimeScript RT Master Mix. Time, temperature, and procedures were strictly followed according to the manufacturer's instructions. Real-time quantitative reverse transcription polymerase chain reaction (qRT-PCR) was performed according to the instructions of the 2× ChamQ SYBR qPCR Master Mix. The PCR reaction system consisted of: 7.2 μL ddH$_2$O, 0.4 μL each of forward and reverse primers, 2 μL cDNA, and 6.25 μL 2× ChamQ SYBR qPCR Master Mix. The qPCR protocol was as follows: 95 °C for 3 s, 98 °C for 0.06 s, 56 °C for 30 s (40 cycles), and 95 °C for 10 s. Three biological replicates were included in the experiment. qPCR amplification and detection were carried out on a LightCycler 480 System (Roche).

## Differential gene function annotation and pathway enrichment analysis

Using the Gene Ontology (GO) database (http://geneontology.org), genes can be classified according to the biological process (BP) in which they are involved, the components that make up the cell (cellular component, CC), and the molecular function they fulfil (molecular function, MF). Differentially expressed genes were annotated with GO terms to identify significantly enriched functional categories Using the Kyoto

Encyclopedia of Genes and Genomes (KEGG) database (https://www.genome.jp/kegg/), genes can be classified according to the pathways they participate in or the functions they perform. The differentially expressed genes were subjected to KEGG annotation for pathway analysis. Heat mapping was performed using TB Tools software.

### Weighted gene co-expression network analysis

The WGCNA software (version 1.72) for R (version 4.3.1; *R Core Team, 2023*) was used to construct gene co-expression networks for barley grain RNA-Seq data. The transcript levels of all genes with expression means ≥1 and coefficients of variation ≥0.1 were converted into a similarity matrix. Genes with similar expression patterns were classified into different modules by combining the phenotypic data and using a bottom-up algorithm. The minimum number of genes in each module was set to 30, and the weight value between screening nodes was greater than 0.02. Gene correlations within modules were demonstrated by Cytoscape Gene Regulatory Network Display Software.

### Protein interaction network analysis

Protein Interaction (PPI) network analysis was used to investigate whether gene product-protein interactions exist, based on the protein information corresponding to the genes. Protein interaction network analysis was carried out using the STRING database (http://string-db.org/) for genes with a combined value greater than 0.4 in the module. PPI networks were visualised by Cytoscape software (version 3.9.1). Cytoscape's Analyze Network was used to calculate the degree value of each gene in the module, and genes with a degree value greater than or equal to 10 were considered hub genes.

### Expression patterns of seven hub genes in naked barley

qRT-PCR experiments followed MIQE guidelines (*Bustin, 2024*). Quantitative primers were designed using Primer 5.0 (Table S1). The cDNAs of grains of Shenglibai and Lalu Qingke at early, mid and late filling periods were used as templates, and 18SrRNA was selected as the internal reference gene, and TaKaRa's TB GreenpremixExTaq II fluorescent dye was used for qRT-PCR using LightCycler480System. The qRT-PCR reaction system was performed by referring to *Yao et al. (2021)*. Expression data were collated and analysed using Microsoft Excel 2010 and SPSS 22.0 statistical software, respectively, and presented as mean ± standard deviation (SD). Relative expression of genes in barley varieties at different times was calculated using the $2^{-\Delta\Delta Ct}$ formula (*Pfaffl, 2001*). All experiments included three biological replicates, and the experimental data were analyzed using ANOVA (SPSS 22.0).

### Statistical analysis

All experiments were conducted with three biological replicates, and the experimental data were statistically analyzed through ANOVA using SPSS 22.0, and $P < 0.05$ was considered a statistically signficant difference.

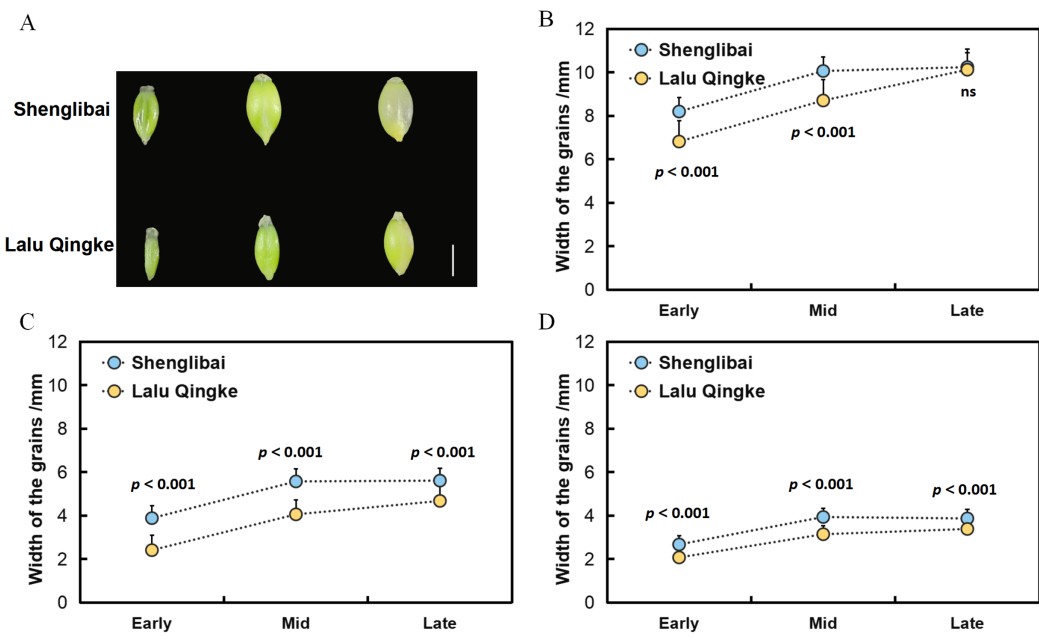

**Figure 1 Grain size dynamics of two varieties during development.** (A) Dynamic changes in grain morphology of Shenglibai and Lalu Qingke at different grain-filling periods. (B) Changes in grain length of the two varieties across three developmental periods. (C) Changes in grain width of the two varieties across three developmental periods. (D) Changes in grain thickness of the two varieties across three developmental periods.

## RESULTS

### Significance analysis of grain phenotypes (GL, GW and GT) of both varieties

To elucidate the mechanisms underlying grain size variation between the two naked barley varieties, we initially conducted morphological observations of their grains across various filling periods (Fig. 1A). Our findings revealed a swift expansion of grains post-filling, marked by substantial increments in GL, GW, and GT (Fig. 1). Midway through the filling phase, these dimensions began to stabilize, maintaining a steady growth trajectory (Fig. 1). Upon reaching the conclusion of the filling period, it was observed that while the GL did not significantly differ between the two varieties, both GW and GT exhibited notable variations throughout all three filling intervals (Fig. 1, Table 1).

### Quality assessment of RNA-seq data

RNA quality assessment is shown in Table S2. The statistical power of this experimental design was calculated using RNASeqPower (parameters: $n = 3$ biological replicates per group, sequencing depth = 16 M reads, effect size = 2-fold, $\alpha = 0.05$). The results demonstrated a power of 94%, indicating reliable detection of expression differences ≥2-fold. Transcriptome sequencing of grain samples from three developmental periods of two naked barley varieties with extreme grain sizes produced a total of 849.56 million of clean

**Table 1 Significance analysis of grain length, width and thickness at three periods for both varieties.**

| Period | Variety | Length | Width | Thickness |
|---|---|---|---|---|
| Early filling period | Shenglibai | $8.19 \pm 0.11^{***}$ | $3.88 \pm 0.01^{***}$ | $2.67 \pm 0.06^{***}$ |
| | Lalu Qingke | $6.81 \pm 0.28^{***}$ | $2.4 \pm 0.08^{***}$ | $2.07 \pm 0.02^{***}$ |
| Mid filling period | Shenglibai | $10.06 \pm 0.01^{***}$ | $5.57 \pm 0.01^{***}$ | $3.93 \pm 0.15^{***}$ |
| | Lalu Qingke | $8.7 \pm 0.18^{***}$ | $4.05 \pm 0.10^{***}$ | $3.14 \pm 0.05^{***}$ |
| Late filling period | Shenglibai | $10.24 \pm 0.13^{ns}$ | $5.6 \pm 0.05^{***}$ | $3.86 \pm 0.09^{***}$ |
| | Lalu Qingke | $10.12 \pm 0.23^{ns}$ | $4.67 \pm 0.02^{***}$ | $3.38 \pm 0.01^{***}$ |

**Notes:**
$^{***}$ $p < 0.001$.
ns: did not achieve statistical significance.

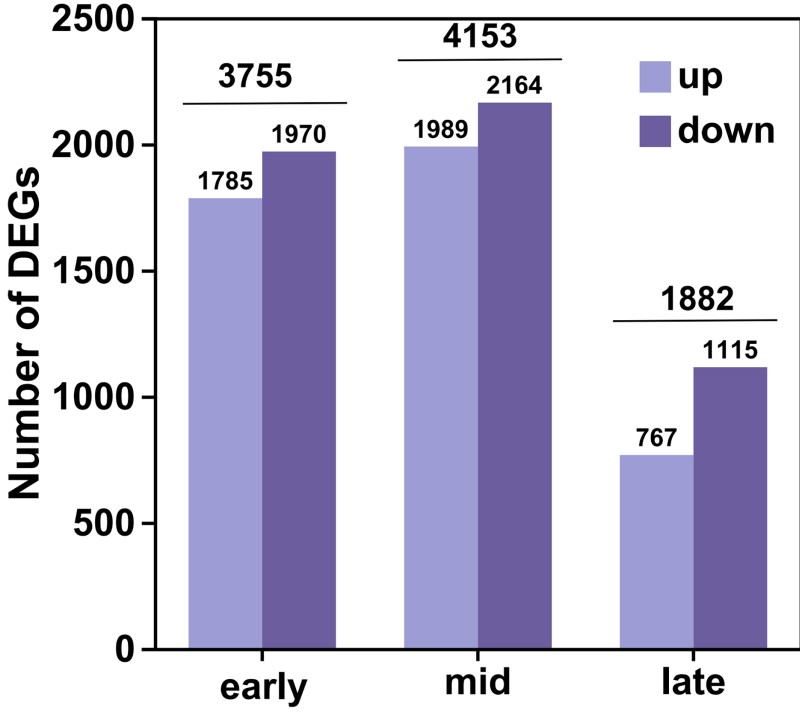

**Figure 2 DEGs statistics.** The horizontal coordinates represent the different groups of differential comparisons and the vertical coordinates represent the corresponding number of up-and down-regulated genes.

reads and 122.17 GB of clean bases (Table S3). The effective data volume for each sample ranged from 6.35 to 7.06 GB, with Q30 base percentages between 94.02% and 95.66%, and an average GC content of 49.87% (Table S3). These metrics indicate that the RNA-seq data were of high quality and suitable for subsequent analyses. By aligning the reads to the barley reference genome (available at http://plants.ensembl.org/Hordeum_vulgare/Info/Index), we achieved genome comparison rates for individual samples ranging from 93.48% to 95.65%.

## Analysis of differences in expression

In this analysis, a total of 35,826 genes were detected, of which 35,106 were known genes (Table S4). Significant differences in gene expression were identified under the criteria of fold change (FC) $\geq$ 2 or FC $\leq$ 0.5 and an adjusted $P$-value ($P$ adjust) < 0.05. This resulted in 4,541 up-regulated genes and 5,249 down-regulated genes, culminating in a total of 6,438 differential genes after de-duplication (Fig. 2). During the early filling period, 1,785 genes were up-regulated and 1,970 were down-regulated. In the mid-irrigation period, 1,989 genes were up-regulated and 2,164 were down-regulated. At the late filling period, 764 genes were up-regulated and 1,115 were down-regulated. The highest number of differential genes was observed during the mid-filling period, while the lowest number was recorded at the late-filling period. This suggests that the differentially expressed genes associated with grain size development primarily begin to regulate their expression during the early and middle periods of filling.

## Weighted correlation network analysis

To investigate the relationship between sample expression profiles and sample phenotypes, a WGCNA was conducted. This method requires the removal of low-expression genes or transcripts with minimal variation, as these typically represent noise and can interfere with the construction of reliable co-expression networks. In this study, 35,826 genes were initially expressed; however, genes with a mean expression level below 1 and a coefficient of variation less than 0.1 were excluded to enhance the quality of the analysis. After this filtering process, a total of 13,079 genes were retained for further investigation. Gene co-expression was assessed using a soft thresholding approach, with a threshold value of $\beta$ = 9 selected for module identification (Fig. 3A). To define the co-expression modules, hierarchical clustering was performed, setting a minimum of 30 genes per module. Modules with a shear height below 0.25 were merged into a single module (Figs. 3B, 3C). Correlations between the identified modules were then calculated using SL, SW, and ST as trait matrices. This process resulted in the identification of five modules, excluding the grey module (Fig. 3D). The number of genes in each module ranged from 75 (green module) to 11,698 (turquoise module). Among these, the blue module showed significant positive correlations with all three traits: SL (r = 0.95, $p$ = 0), SW (r = 0.818, $p$ = 0), and ST (r = 0.798, $p$ = 0).

## Significant enrichment of grain size-related pathways in the blue module

The Blue module comprises 667 genes, and when these were intersected with 6,438 DEGs, a total of 587 DEGs were identified (Table S5). To elucidate the functions of these 587 DEGs, we conducted a GO enrichment analysis. The analysis revealed 279 GO terms, which included 162 related to BP, 21 associated with CC, and 96 pertaining to MF (Fig. 4A, Table S6). Within the category of BP, the term 'response to oxygen-containing compound' was the most prominent, encompassing 22 genes. Regarding cellular components, the 'extracellular region' exhibited the highest enrichment score, involving 36 genes. For molecular functions, the term 'hydrolase activity, acting on glycosyl bonds' achieved the
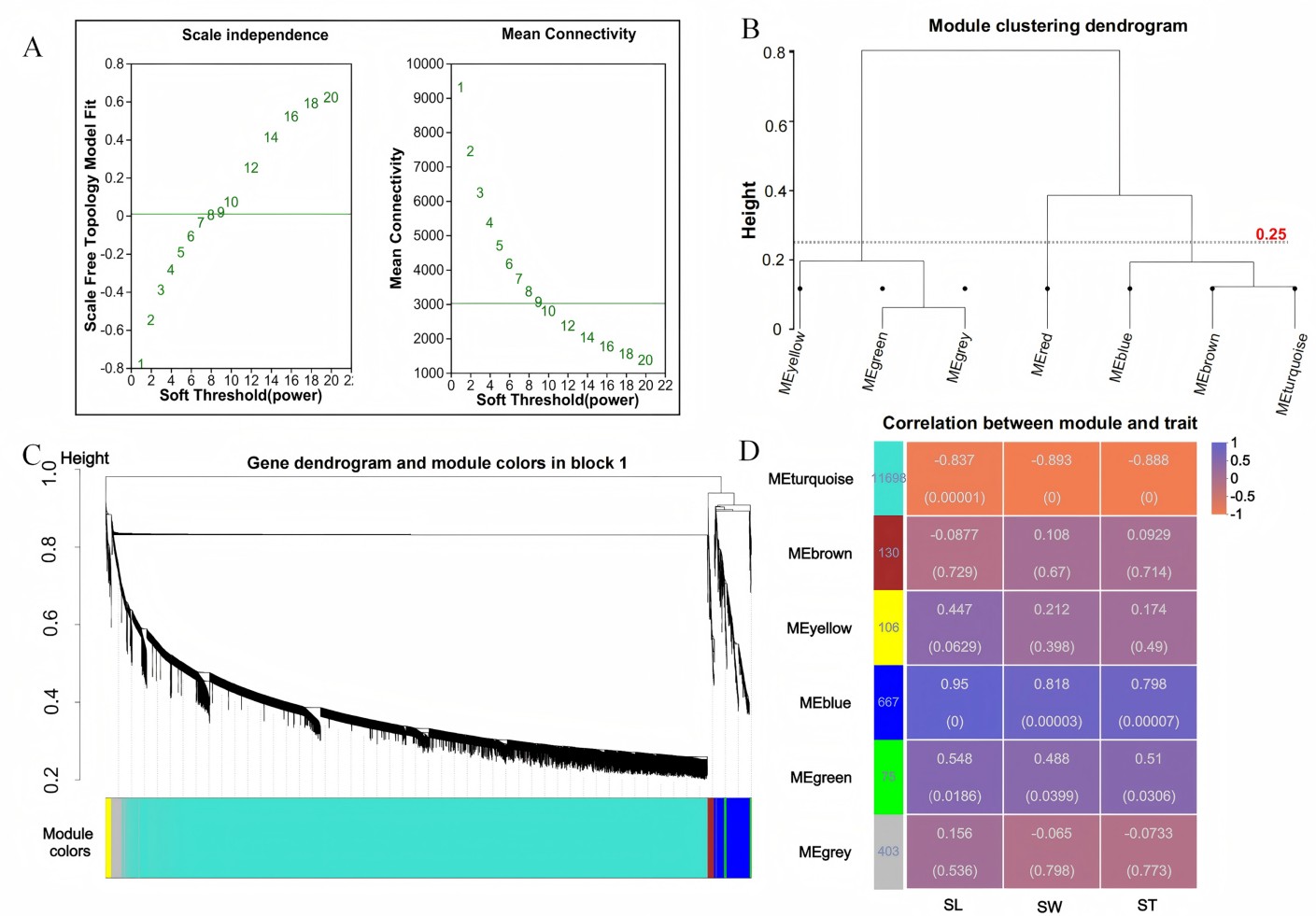

**Figure 3 WGCNA analysis.** (A) Scale-free topology fitting curve and mean connectivity curve. The horizontal axis represents the soft-thresholding power β, the vertical axis represents the scale-free topology model fit ($R^2$) corresponding to the adjacency matrix transformed under the given β value. A higher $R^2$ indicates that the network better approximates a scale-free distribution (left figure); The horizontal axis represents the soft-thresholding power β; the vertical axis represents the average connectivity (k/degree) of each node in the network corresponding to the adjacency matrix transformed under the given β value (right figure). (B) Module clustering dendrogram. Branches represent individual modules. The closer two module branches are, the more similar the modules, which can serve as a basis for module merging. The vertical axis represents the clustering distance. (C) Module classification tree. Genes are divided into modules based on their expression trends. Each branch represents a gene, and each color represents a module. Genes colored in gray are those not assigned to any specific module. (D) Module-trait relationship heatmap. The heatmap illustrates the correlation between modules and specific traits. The horizontal axis represents different traits, and the vertical axis represents different modules. The numbers in the leftmost column indicate the number of genes in each module. The data on the right side of each cell shows the correlation coefficient between the module and the trait, along with the significance *P*-value (in parentheses). The larger the absolute value, the stronger the correlation. Orange indicates a negative correlation, while purple indicates a positive correlation.

highest score, with 24 genes represented. The KEGG database was utilized to analyze and further elucidate the biological functions and interactions of the identified genes (Fig. 4B, Table S7). Among the KEGG annotation entries, the three most enriched pathways were 'Cyanoamino acid metabolism', 'Biosynthesis of various plant secondary metabolites', and 'Starch and sucrose metabolism', all of which were significantly associated with the selected traits. Notably, the starch and sucrose metabolism pathway contained the highest number

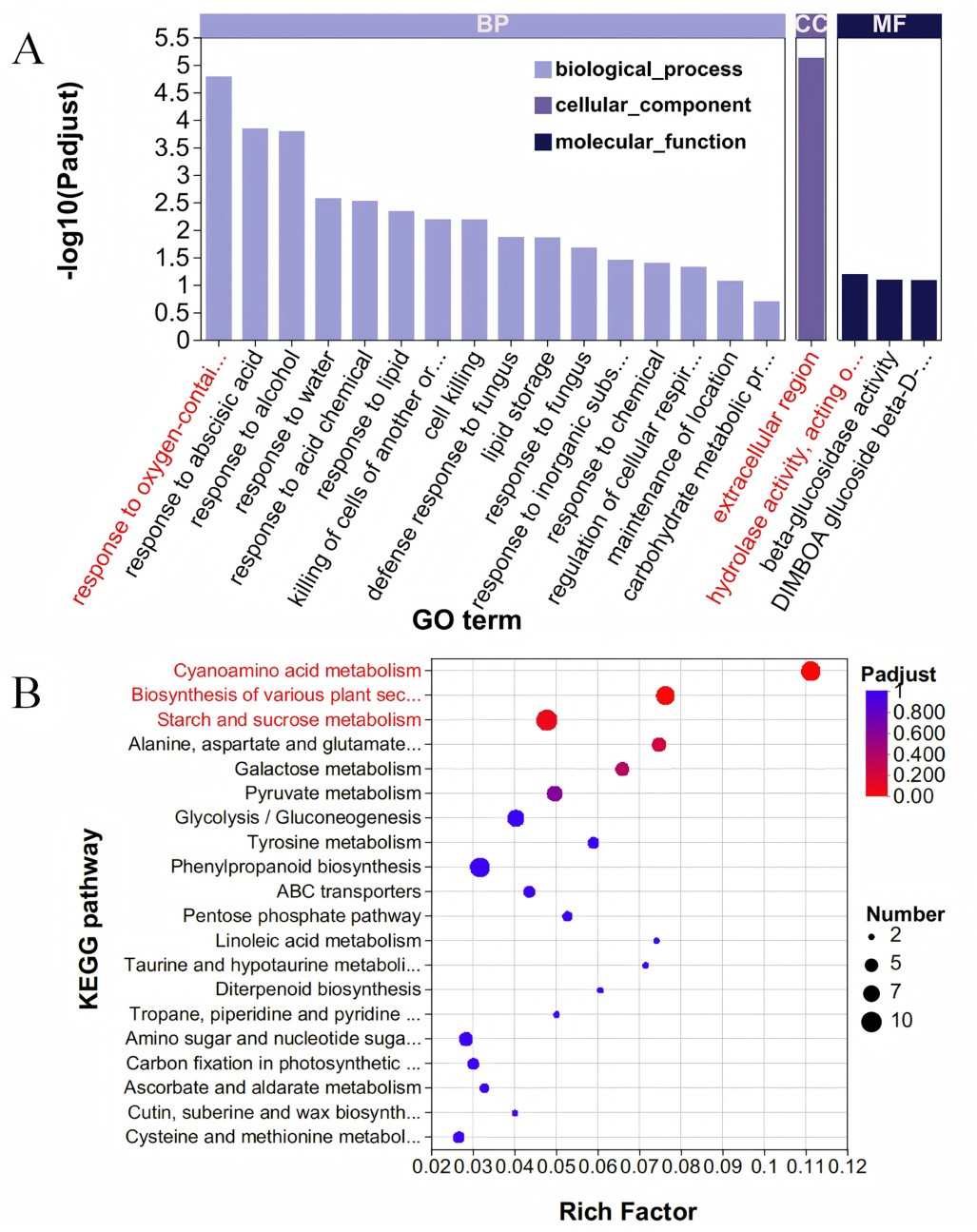

**Figure 4 GO and KEGG analysis.** (A) GO enrichment analysis. The vertical axis represents GO terms, and the horizontal axis represents the Rich factor. A larger Rich factor indicates a higher degree of enrichment. The size of the points represents the number of genes in the corresponding GO term, and the color of the points corresponds to different ranges of adjusted *P*-values (padjust). (B) KEGG enrichment analysis. The vertical axis represents pathway names, and the horizontal axis represents the ratio of Rich factor to the number of annotated genes (background number). A larger Rich factor indicates a higher degree of enrichment. The size of the points represents the number of genes in the corresponding pathway, while the color of the points corresponds to different ranges of adjusted *P*-values (padjust).

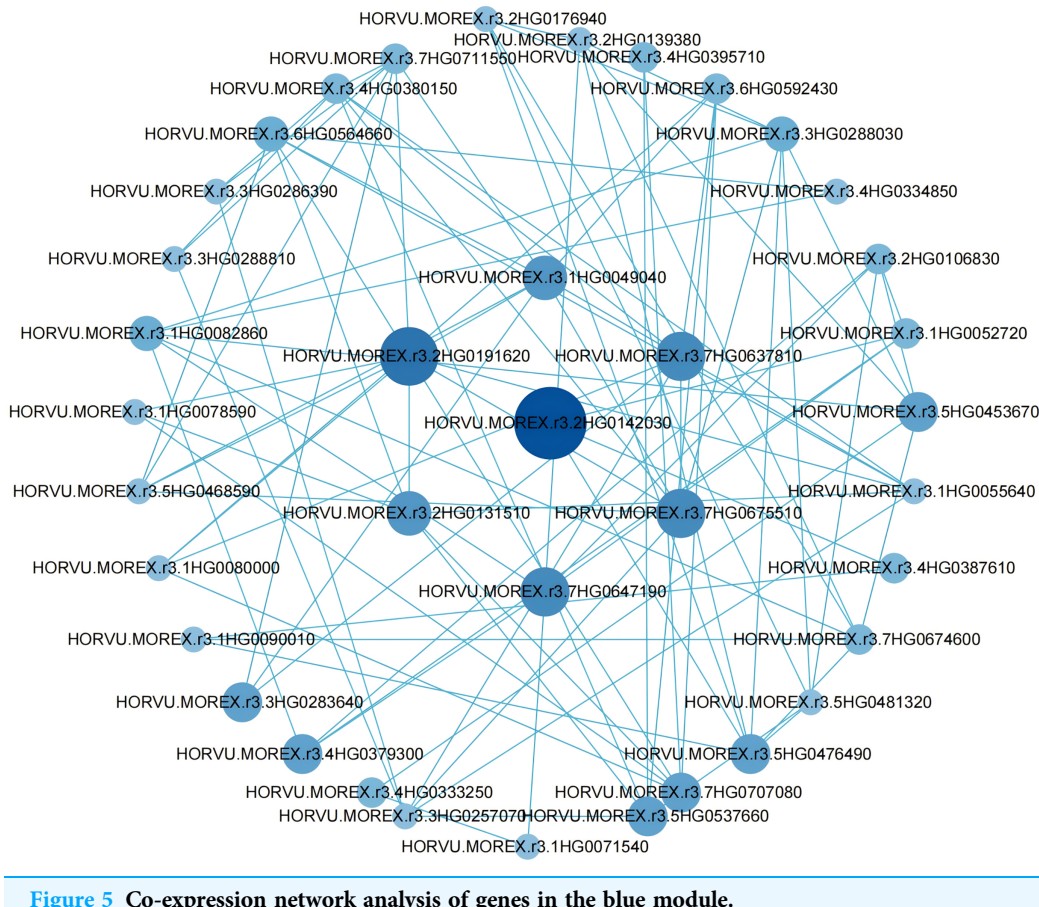

**Figure 5 Co-expression network analysis of genes in the blue module.**

of genes, with a total of 10 genes, followed by the cyanoamino acid metabolism pathway with nine genes, and the biosynthesis of various plant secondary metabolites pathway with eight genes.

## Acquisition of hub genes and protein interaction network analysis

To further pinpoint hub genes, a PPI network was constructed using differential genes with degree values greater than 5 in the blue module (Fig. 5). Clearly, a high correlation between hub genes was observed in the network, suggesting that they are synergistically involved in regulating grain size. Among these genes, we considered seven genes with degree values greater than 8 to be further considered as hub genes (Table S8).

## Evidence of a correlation between predictive genes and grain size

A preliminary analysis of the expression patterns of the seven hub genes was conducted based on transcriptomic data. As illustrated in the figure, these genes generally exhibited an upward trend in expression levels as the grain-filling period progressed. This trend was particularly pronounced in the large-grained varieties (Fig. 6A). To validate the reliability of the transcriptomic data, we performed qRT-PCR confirmation of expression patterns

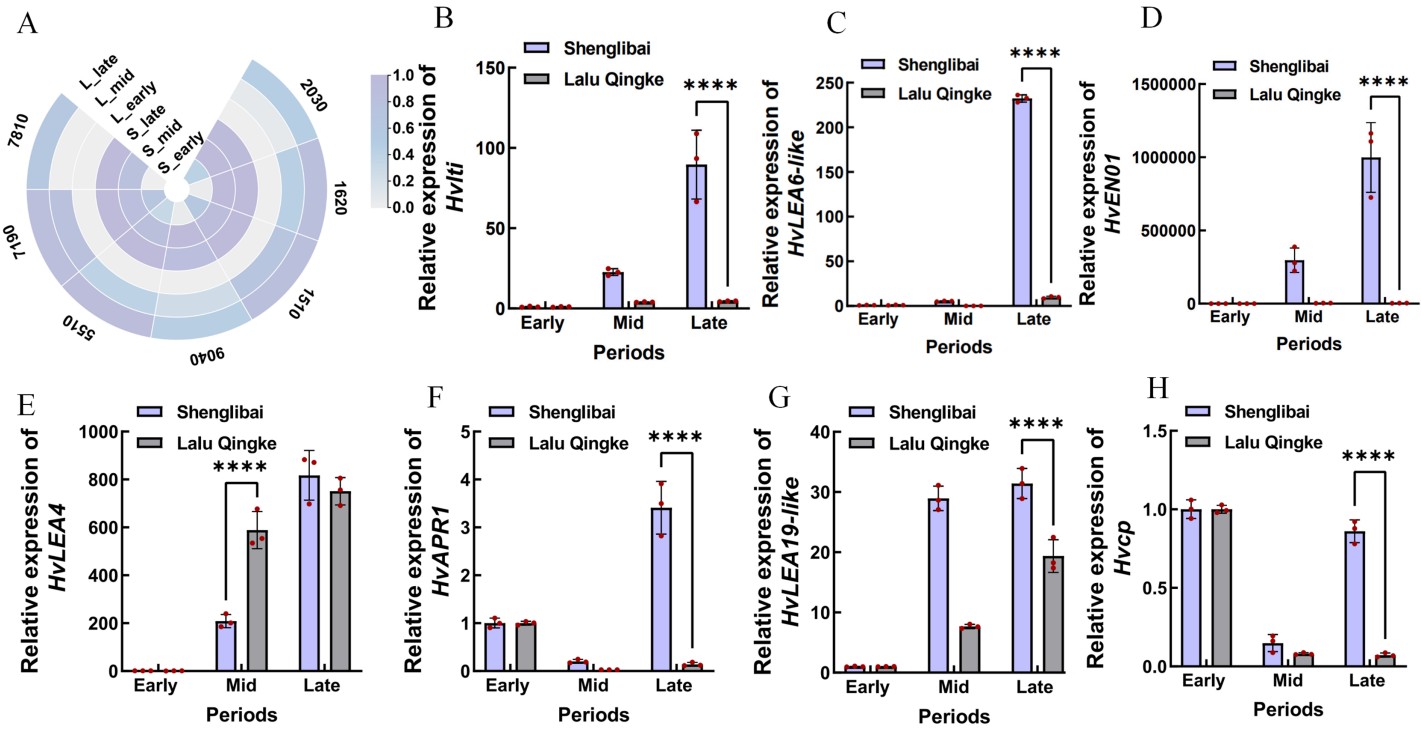

**Figure 6 Expression analysis of seven hub genes.** (A) Expression analysis of seven hub genes in transcriptomic data. S: Shenglibai; L: Lalu Qingke. 7810: HORVU.MOREX.r3.7HG0637810; 9040: HORVU.MOREX.r3.1HG0049040; 7190: HORVU.MOREX.r3.7HG0647190; 5510: HORVU. MOREX.r3.7HG0675510; 2030: HORVU.MOREX.r3.2HG0142030; 1620: HORVU.MOREX.r3.2HG0191620; 1510: HORVU.MOREX.r3.2HG0131 510. (B) HORVU.MOREX.r3.1HG0049040. (C) HORVU.MOREX.r3.7HG0637810. (D) HORVU.MOREX.r3.7HG0647190. (E) HORVU.MOREX. r3.7HG0675510. (F) HORVU.MOREX.r3.2HG0142030. (G) HORVU.MOREX.r3.2HG0191620. (H) HORVU.MOREX.r3.2HG0131510. Asterisks (****) indicate a statistically significant difference ($p < 0.0001$).

for seven hub genes. Figure 6 illustrates period-specific expression dynamics of these candidate genes across three grain-filling phases (early, middle, and late) in two contrasting varieties. The qRT-PCR results showed strong concordance with transcriptomic profiling data, particularly in varietal expression trends. Notable examples include *Hvlti* (HORVU.MOREX.r3.1HG0049040), which exhibited pronounced mid-phase upregulation in the large-grain variety (21.98-fold by qRT-PCR *vs*. 15.9-fold RNA-Seq), contrasting with its modest induction in the small-grain counterpart (3.95-fold *vs*. 2.59-fold). *HvLEA6-like* (HORVU.MOREX.r3.7HG0637810) showed distinct phasic expression - reaching peak levels during middle and late phases in large grains while being transcriptionally silent (undetectable) in small grains during mid-phase, suggesting its potential regulatory role in cellular proliferation during grain development. The most striking varietal divergence appeared in *HvENO1* (HORVU.MOREX.r3.7HG0647190) expression dynamics. Both varieties maintained basal expression during early filling, but the large-grain type displayed dramatic mid-phase amplification (82.11-fold higher than small-grain) that sustained through late phase (278.75-fold differential), coinciding with phenotypic divergence in grain dimensional expansion (GW/GT). This expression-surge/decline pattern mirrored the contrasting morphological trajectories between varieties.

*HvLEA4* (HORVU.MOREX.r3.7HG0675510) demonstrated progressive accumulation in both varieties, yet paradoxically showed significantly elevated mid-phase expression in small grains. Conversely, three late-phase regulators-*HvAPR1* (HORVU.MOREX. r3.2HG0142030), *HvLEA19-like* (HORVU.MOREX.r3.2HG0191620), and *Hvcp* (HORVU.MOREX.r3.2HG0131510) - exhibited marked large-grain preferential expression during terminal filling periods.

## DISCUSSION

Grain size is a crucial indicator of crop yield. Investigating the key modules and genes related to grain size in naked barley will enhance our understanding of its molecular mechanisms and significantly benefit molecular breeding efforts. The integration of phenotypic, transcriptomic, and co-expression network analyses in this study provides critical insights into the molecular mechanisms governing grain size variation between two naked barley varieties. Monitoring of phenotypic dynamics revealed significant differentiation in GW and GT among the tested varieties during the pre-, mid-, and late periods of filling ($P < 0.001$). In contrast, differences in GL were significant only during the pre-and mid-periods ($P < 0.001$) and became non-significant at the late period (Table 1). These findings collectively suggest that the radial expansion parameter plays a crucial role in determining the final grain morphology in both varieties.

The high-resolution transcriptome data (Q30 > 93%, mapping rate > 93.48%) not only provided a robust foundation for subsequent differential expression analysis and functional annotation but also established a reliable basis for weighted gene WGCNA. Differential expression analysis uncovered dynamic changes in gene expression across various developmental periods of naked barley grains. We observed that the number of DEGs peaked during the mid-grain filling period and declined during the late-grain filling period. This indicates that the critical regulatory phase for grain size development in naked barley likely occurs predominantly during the mid-grain filling period. Grain development is a dynamic process encompassing three key phases: the initial grain formation period, the middle dry matter accumulation period, and the maturation period (*Du et al., 2019*). During the middle period, grain weight increases almost linearly (*Du et al., 2019*), which may explain the higher number of DEGs observed during this period, likely driven by the synthesis of substantial dry matter. Notably, some of the DEGs identified in this study have been previously linked to grain size regulation in rice and maize. For instance, the *CBL* gene has been reported to enhance grain size by promoting cellular expansion in rice glumes (*Zhang et al., 2023b*), while overexpression of the *SUS1* gene in maize has been shown to increase straight-chain starch content and grain size (*Li et al., 2023*). These findings further validate the reliability of our results and highlight potential target genes for future research on grain size regulation.

WGCNA unveiled a robust co-expression network, with the blue module exhibiting strong positive correlations with GL, GW, and GT. This module comprises 667 genes significantly enriched in pathways associated with grain development. GO enrichment analysis identified key functional categories, including "response to oxygen-containing

compounds," "extracellular region," and "hydrolase activity, acting on glycosyl bonds," indicating that oxidative stress response, extracellular signalling, and carbohydrate metabolism are integral to grain development (*MacNeill et al., 2017*). KEGG analysis further highlighted the importance of pathways such as starch and sucrose metabolism, and biosynthesis of plant secondary metabolites. These pathways are crucial for grain growth and development, as they regulate carbon allocation, energy metabolism, and the synthesis of structural and protective compounds (*Chen et al., 2023*; *Böttger et al., 2018*; *Kumar et al., 2023*). The strong correlation between the blue module and grain size traits suggests that the genes within this module play a pivotal role in determining grain morphology.

The PPI network analysis identified seven hub genes with high connectivity, indicating their central role in the regulatory network governing grain size. These genes—*Hvcp, Hvlti, HvENO1, HvLEA4, HvAPR1, HvLEA19-like, and HvLEA6-like*—exhibited period-specific and variety-specific expression patterns during grain filling. *Hvcp* encodes cysteine protease RD19D, which regulates the lipid catabolic process (*Ye et al., 2020*). Additionally, studies have shown that cysteine protease RD19-like is associated with disease resistance in wheat (*Shi et al., 2021*), suggesting that *Hvcp* may have similar functions. Unfortunately, there are currently no reports on the role of this type of enzyme in regulating grain size or weight. Similarly, the gene *Hvlti*, which encodes a low-temperature-induced protein, has not been reported to be associated with grain size in other species. Notably, *HvENO1* displayed dramatic upregulation during the mid- and late-grain filling periods in large-grain varieties, coinciding with significant phenotypic divergence in grain size. This gene encodes enolase, a metalloenzyme that catalyses the dehydration of substrates in the glycolytic pathway and is presumably essential for cell expansion during grain development (*Zhu & McBride, 2021*). The role of *ENO* genes in regulating organ size has been previously reported; for instance, in tomato, *ENO* regulates fruit size through the floral meristem developmental network (*Yuste-Lisbona et al., 2020*). Similarly, in Arabidopsis, *ENO2* influences grain size and weight by modulating cytokinin levels and forming the ENO2-bZIP75 complex (*Liu et al., 2020*). It was also found that PROTEIN L-ISOASPARTYL METHYLTRANSFERASE (PIMT) affects grain size and weight by regulating enolase (ENO) activity (*Kamble et al., 2024*). Among the seven hub genes, three encode late embryogenesis abundant (LEA) proteins, which are hypothesized to play roles in cellular protection and desiccation tolerance during grain maturation (*Huang et al., 2022*; *Jia et al., 2022*; *Zhang et al., 2023a*). Research shows that silencing the *LEA* gene in naked barley significantly impairs drought tolerance (*Liang et al., 2012*). The *LEA* gene in barley confers dehydration tolerance to transgenic rice by protecting cell membranes (*Babu et al., 2004*). These findings highlight the functional diversity of the identified hub genes and their potential contributions to grain size regulation through distinct molecular mechanisms.

The qRT-PCR validation of the expression patterns of the central genes not only confirmed the reliability of the transcriptomic data but also provided robust evidence for the functional relevance of these genes in grain size regulation. In large-grained varieties,

*Hvlti*, *HvENO1*, and *HvLEA6-like* exhibited significant up-regulation during the critical periods of grain filling, underscoring their potential as pivotal regulators of grain size. Notably, *HvENO1* displayed an extraordinary expression pattern, with its transcript levels in large grains being tens to hundreds of times higher during the middle and late periods of grain filling compared to small grains. This dramatic differential expression coincided with the phenotypic divergence in grain size, suggesting that *HvENO1* may play a crucial role in driving cellular expansion and nutrient accumulation during grain development. Similarly, *Hvlti* and *HvLEA6-like* also showed pronounced up-regulation in large-grained varieties, further supporting their involvement in grain size determination. Given the striking expression dynamics and their potential regulatory roles, we plan to conduct a series of functional studies focused on these genes. These studies will include gene knockout or knockdown experiments to elucidate their precise roles in grain development, as well as overexpression assays to explore their potential for enhancing grain size and yield. Additionally, we aim to investigate the molecular mechanisms underlying their regulation, including potential interactions with other genes and pathways involved in grain filling and size determination. Unraveling the functional roles of *HvLti*, *HvENO1*, and *HvLEA6-like* genes is crucial for elucidating their molecular regulatory networks. Translating these findings into functional markers for marker-assisted selection (MAS) offers the potential to dramatically expedite the breeding of elite, high-yield barley varieties with desirable large-grain phenotypes.

## CONCLUSION

This study explored the molecular basis of grain size variation in barley through phenotypic and transcriptomic analyses. Phenotypic data showed significant differences in GW and GT during mid-grain filling, while RNA-seq identified 6,438 DEGs, peaking in mid-filling. Weighted WGCNA revealed a blue module strongly linked to grain size, enriched in starch/sucrose metabolism and related pathways. Seven hub genes, including *HvENO1* and *HvLEA4*, were identified as key regulators, with *HvENO1* showing a 278.75-fold expression difference in large-grain varieties during mid- and late filling. qRT-PCR validation confirmed their period- and variety-specific expression, aligning with phenotypic traits. These findings highlight critical genetic and metabolic regulators of grain size, offering targets for barley breeding.

## ABBREVIATIONS

**WGCNA**     Weighted gene co-expression network analysis
**RNA-Seq**   transcriptomic
**qRT-PCR**   Quantitative reverse transcription polymerase chain reaction
**DEGs**      Differentially expressed genes
**GL**        Grain length
**GW**        Grain width
**GT**        Grain thickness

### Funding

This research was supported by the Qinghai University Special Project for Enhancing Scientific Research Capabilities (2025KTST09), the China Association for Science and Technology (CAST) Youth Talent Support Program for Doctoral Students, Qinghai Province Joint Research Initiative for Highland Barley Breeding, and the Agriculture Research System of China (CARS-05). The funders had no role in study design, data collection and analysis, decision to publish, or preparation of the manuscript.

### Grant Disclosures

The following grant information was disclosed by the authors:
Qinghai University Special Project for Enhancing Scientific Research Capabilities: 2025KTST09.
China Association for Science and Technology (CAST).
Youth Talent Support Program for Doctoral Students.
Qinghai Province Joint Research Initiative for Highland Barley Breeding.
Agriculture Research System of China (CARS-05).

### Competing Interests

The authors declare that they have no competing interests.

### Author Contributions

- Yan Wang conceived and designed the experiments, performed the experiments, analyzed the data, prepared figures and/or tables, and approved the final draft.
- Jiahao Zhou performed the experiments, analyzed the data, prepared figures and/or tables, and approved the final draft.
- Mingqi Yang performed the experiments, analyzed the data, prepared figures and/or tables, and approved the final draft.
- Youhua Yao analyzed the data, authored or reviewed drafts of the article, and approved the final draft.
- Yongmei Cui analyzed the data, authored or reviewed drafts of the article, and approved the final draft.
- Xin Li analyzed the data, prepared figures and/or tables, authored or reviewed drafts of the article, and approved the final draft.
- Baojun Ding analyzed the data, prepared figures and/or tables, authored or reviewed drafts of the article, and approved the final draft.
- Xiaohua Yao conceived and designed the experiments, analyzed the data, prepared figures and/or tables, authored or reviewed drafts of the article, and approved the final draft.
- Kunlun Wu conceived and designed the experiments, analyzed the data, authored or reviewed drafts of the article, and approved the final draft.

## Data Availability

Raw data is available in the Supplemental Files and at the National Center for Biotechnology Information (NCBI): PRJNA1232405.

## Supplemental Information

Supplemental information for this article can be found online at http://dx.doi.org/10.7717/peerj.19856#supplemental-information.

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
