# Peer review of "Transcriptomic insights into grain size development in naked barley (Hordeum vulgare L. var. nudum Hook. f): based on weighted gene co-expression network analysis"

_PeerJ, doi:10.7717/peerj.19856_

## Round 0.1 · original submission · Minor Revisions

We have received feedback from the reviewers. Your manuscript requires minor revisions as suggested. Please review the comments and incorporate the necessary changes.

Reviewer 1 ·

Basic reporting

• The manuscript is written in clear, professional scientific English. However, occasional minor grammatical errors or awkward phrasing (e.g., "grouting phase" instead of "filling phase") should be corrected.
• The abstract covers all the essential elements and describes the methodology and the main finding properly. However, it is Slightly dense in places due to technical terms stacked together that could break into shorter sentences or rephrase for improved readability. The final sentence lists gene validation but could more clearly highlight why this matters (the impact). May add a conclusive phrase: “…further supporting the potential of these genes as targets for improving grain size in breeding.”
• The structure is conventional and appropriate, with logical progression from introduction through methods, results, discussion, and conclusion.
• Figures are well-labelled, of good quality, and relevant.
• The introduction establishes a solid context for the research and reflects adequate literature coverage.
• Recent and relevant references are included, especially in discussing known grain size-related genes and mechanisms from rice and maize.
• Some citations could be further clarified to improve their relevance (e.g., LEA proteins or CBL pathway discussion could reference more barley-specific work if available).
• High-quality RNA-Seq data (Q30 > 94%, GC ~50%, mapping rate >93%) ensures reliable transcriptome profiling.
• Identification of 6,438 DEGs and their correlation with phenotypic differences is well justified.
• Results are presented logically, and major findings (e.g., blue module from WGCNA correlating with grain width and thickness) are well supported by data.
• The use of both bioinformatics and experimental validation (qRT-PCR) reinforces the credibility of the findings.
• The discussion is comprehensive and thoughtful, connecting findings to known molecular pathways in rice, maize, and barley.
• Findings are discussed in the context of recent research (2022–2024), and appropriate comparative insights are provided.
However, there are a few suggestions for further improvement.
Revisions Suggested:
• While the discussion references LEA and ENO proteins from other crops, further functional validation in barley/qingke is needed and acknowledged as future work.
• Authors may consider adding more functional genomics perspectives or predictions for marker-assisted selection.
• Minor language polishing for clarity and consistency.
• Clarify certain biological terms (e.g., replace “grouting phase” with “filling phase”).
• If possible, improve the specificity of some references in the LEA/CBL gene discussion with barley-centric studies.
• Consider including an outlook section on how these findings could be applied in breeding programs.
Here is a line wise editorial correction
Line 29: "Phenotypic assessments revealed that grains underwent rapid expansion post-filling..."
➤ "post-filling" is unclear; use "during the filling period".
Line 75: “Kernel tissues were three developmental milestones �”
➤ Fix encoding: replace “�” with "—" or appropriate punctuation.
Line 88–90: “Plant materials” — repetitive phrasing; streamline to:
➤ “The large- and small-grain qingke varieties Shenglibai and Lalu Qingke, respectively, were sourced from the National Crop Germplasm Bank.”
Line 171: “statistically analyzed through ANOVA”
➤ Suggestion: "analyzed using ANOVA (SPSS 22.0)".
Line 179: “grouting phase”
➤ Replace all instances with “filling phase” (common in cereal development literature).
Line 245–249: Long list of HORVU IDs
➤ Suggestion: refer to gene symbols in main text and provide full HORVU IDs in a supplementary table instead.
Inconsistencies
• Use of “early/mid/late filling period” vs. “stage 1/2/3” is inconsistent. Standardize across text and figures.
• Some parts of the manuscript use GL, GW, GT, while others spell out length/width/thickness. Define and maintain consistency.
FIGURE & TABLE ANALYSIS
Figure 1: Grain size dynamics of two varieties during development
Subfigures (A–D) show morphological differences and dimensions (length, width, thickness) across three developmental stages. However, the authors should include
1. *error bars and significance indicators (e.g., p < 0.05) on bar plots in B–D.
2. Add precise y-axis labels with units (e.g., mm).
3. Label developmental stages (1, 2, 3) more explicitly in the figure or legend (early, mid, late).
Figure 2: DEGs statistics
1. Clear and functional, visually shows where transcriptional shifts are strongest (mid filling period).
2. Consider adding total DEG counts at the top of each bar for clarity.
Figure 3: WGCNA analysis
1. Panel C (module tree) would benefit from larger font size or improved resolution.
2. Panel D (heatmap) should include a color scale legend (to indicate strength of correlation).
Figure 4: GO and KEGG analysis
1. Clarify axes titles and enrich color legend explanation (padj values).
2. Annotate the top 3 terms in each panel (BP, CC, MF; or KEGG pathways).
3. Consider using horizontal orientation for long GO term names.
Figure 5: Co-expression network of blue module genes
Increase resolution/clarity; nodes are hard to distinguish.
Figure 6: Expression analysis of 7 hub genes
1. Panels B–H (qRT-PCR) could benefit from statistical significance annotations and error bars.
2. Consider combining panels into a compact grid layout.
3. Define all axes clearly (e.g., expression relative to what? Ct fold?).
Table 1: Grain size statistics across three stages
1. Provide mean ± SD explicitly for each value.
2. Move significance stars (***, ns) into footnotes or add to each numeric value for clarity.
3. Add horizontal borders for better table readability.

Experimental design

High-quality RNA-Seq data (Q30 > 94%, GC ~50%, mapping rate >93%) ensures reliable transcriptome profiling.
• Identification of 6,438 DEGs and their correlation with phenotypic differences is well justified.

Validity of the findings

Results are presented logically, and major findings (e.g., blue module from WGCNA correlating with grain width and thickness) are well supported by data.
• The use of both bioinformatics and experimental validation (qRT-PCR) reinforces the credibility of the findings.
• The discussion is comprehensive and thoughtful, connecting findings to known molecular pathways in rice, maize, and barley.
• Findings are discussed in the context of recent research (2022–2024), and appropriate comparative insights are provided.

Additional comments

Overall quality of the research is good and I recommend this manuscript for publication as the study addresses a biologically important and understudied system: grain size regulation in qingke, a high-altitude barley variant and It provides first-of-its-kind transcriptomic insights across key developmental stages using RNA-Seq + WGCNA, targeting traits of agronomic and breeding relevance.

Please go through the attached document for the revisions suggested and improve the manuscript accordingly.

·

Basic reporting

The manuscript is clearly written in unambigous english language.

The study reports a detatiled transcriptome analysis of large and small grains naked barley varieties to identify key genes underpinning grain filliong and grain size. The relevent literature is sufficiently cited and provides relevant background of the study.

Overall the structure and organization of the manuscript professional and meets the standard of the journal. To further improve the quality of the manuscript I suggest the following changes

1. The authors are using the term qingke for Tibetan naked barley. This is local term and can be mentioned only onc in the introduction. In over all manuscript, I suggest to use the term that can be understood by international readers. Better use the term naked barley or hulless barley.

2. Line 63-67. Rewrite thhe following lines and setablish connection with background. Do the authors want to write that they used translational genomics to reaveal grain size related signaling pathways in naked barly using the information available in model plants genomics? "Current research has
64 also revealed some regulatory mechanisms and candidate genes related to barley grain size. For
65 example, genes controlling row type (such as Vrs genes), hulled/naked caryopsis (such as Nud
66 genes), and dwarfism (such as sdw/denso genes) have been identified (Youssef et al., 2017;
67 Koppolu et al., 2013; Bull et al., 2017; Taketa et al., 2008; Franckowiak and Lundqvist. 2017)"

3. The introduction can be improved by adding exsiting knowledge on grain size related genes/pathways in naked barley
4. Rewrite the following lines to improve the clarity "Kernel tissues were three developmental milestones ñ initial cell proliferation phase (early filling period), expansion period (mid-filling period), and maturation period (late filling period) ñ representing distinct phases of dimensional determination."
5. The language used in the following sentencs is ambigous.
"Through WGCNA integrating longitudinal phenotypic measurements (length/width/thickness trajectories) with temporal expression profiles, we deciphered gene modules co-varying with morphological expansion. Hub genes within these networks were prioritized through topological centrality metrics, differential expression filtering, and functional enrichment mapping"

6. Line 85- 87. How does gene inventory provides diagnostic markers? Please explain. I dont' see any diagnostic marker reported in the manuscript.

Experimental design

The experimental design is well structured, relevant and meaningful. I appreciate researchers for performing this rigorous study to pinpoint grain size related genes in naked barley. Some minor errors need to be corrected in the revised version.
1. Line 116. Change the 'is' to 'was'. Please past passive voice tense to write material and method section.
2. Please also coorect error in line 128.

3. Line 139. according to the pathway pathways they. Remove repitiion
4. Line 140. 'KEGG annotation of differentially expressed genes' is an incomplete sentence
5. Line 141. Write asv TB Tools

6. Line 173. Change signifi-cant to signficant

Validity of the findings

The finding of the study through high quality transcriptome data. Seven hub genes from the blue module strongly associated with grain size were further confirmed with qRT_PCR. The methods used to analysis transcritome analysis are robust and reliable.
Raw data have been deposited to National Center for Biotechnology Information (NCBI)
404 under the BioProject number PRJNA1232405 405 (https://dataview.ncbi.nlm.nih.gov/object/PRJNA1232405?reviewer=5epc2u659omr96evi6h4d5
406 bnds).

---

## Round 0.2 · accepted · Accept

Following the revisions, the article has met the journal’s scientific and editorial standards and is hereby accepted for publication.

·

Basic reporting

The authors have substantially improved the revised Manuscript according to my comments.

Experimental design

Satisfactory

Validity of the findings

The comments are addressed in the revised Manuscript